# Assessing the Clinical Efficacy of Therapeutic Drug Monitoring for Risperidone and Paliperidone in Patients with Schizophrenia: Insights from a Clinical Data Warehouse

**DOI:** 10.3390/ph17070882

**Published:** 2024-07-03

**Authors:** Wonsuk Shin, Dong Hyeon Lee, Hyounggyoon Yoo, Huiyoung Jung, Minji Bang, Anhye Kim

**Affiliations:** 1Department of Clinical Pharmacology and Therapeutics, CHA Bundang Medical Center, School of Medicine, CHA University, Seongnam-si 13496, Republic of Korea; wonsug89@chamc.co.kr (W.S.); hgyoo0317@cha.ac.kr (H.Y.); huiyjung00@gmail.com (H.J.); 2Department of Physiology, School of Medicine, CHA University, Seongnam-si 13488, Republic of Korea; leedh@cha.ac.kr; 3Department of Pharmacology, School of Medicine, CHA University, Seongnam-si 13488, Republic of Korea; 4Department of Psychiatry, CHA Bundang Medical Center, School of Medicine, CHA University, Seongnam-si 13496, Republic of Korea

**Keywords:** therapeutic drug monitoring, risperidone, paliperidone, schizophrenia, clinical data warehouse, treatment outcomes, antipsychotic medications, psychiatric care, real-world data, adverse effects

## Abstract

This study investigated the usage patterns and impact of therapeutic drug monitoring (TDM) for risperidone and paliperidone in patients diagnosed with schizophrenia, utilizing retrospective real-world data sourced from a single center’s Clinical Data Warehouse. Our study cohort comprised patients diagnosed with schizophrenia undergoing treatment with either risperidone or paliperidone. Data on demographic characteristics, comorbidities, medication utilization, and clinical outcomes were collected. Patients were categorized into two groups: those undergoing TDM and those not undergoing TDM. Additionally, within the TDM group, patients were further stratified based on their risperidone and paliperidone concentrations relative to the reference range. The findings revealed that patients in the TDM group received higher risperidone and paliperidone doses (320 mg/day and 252 mg/day, *p* = 0.0045) compared to their non-TDM counterparts. Nevertheless, no significant disparities were observed in hospitalization rates, duration of hospital stays, or compliance between the two groups (*p* = 0.9082, 0.5861, 0.7516, respectively). Subgroup analysis within the TDM cohort exhibited no notable distinctions in clinical outcomes between patients with concentrations within or surpassing the reference range. Despite the possibility of a selection bias in assigning patients to the groups, this study provides a comprehensive analysis of TDM utilization and its ramifications on schizophrenia treatment outcomes.

## 1. Introduction

Therapeutic drug monitoring (TDM) of antipsychotics is being increasingly recognized as a valuable practice in psychiatric care [1]. It empowers clinicians to optimize treatment by tailoring medication dosages to individual patients, factoring in metabolism, age, weight, smoking habits, concurrent medications, adverse effects, and pharmacokinetically relevant comorbidities (such as hepatic or renal insufficiency and cardiovascular disease), as well as other factors [2]. By ensuring drug levels remain within the therapeutic threshold, TDM assists in minimizing adverse effects, curtails treatment failures, and enhances overall outcomes. Moreover, TDM facilitates assessment of patient adherence to medication regimens and mitigates the risk of drug interactions. Despite incurring additional costs, TDM provides cost-effective overtime, averting hospitalizations and other healthcare utilization linked to treatment failures [3]. Additionally, TDM data contributes significantly to research efforts aimed at comprehending drug pharmacokinetics, pharmacodynamics, and individual responses to treatment. Collectively, TDM assumes a crucial role in optimizing the efficacy, safety, and cost-effectiveness of antipsychotic therapy [4].

Risperidone and paliperidone represent atypical antipsychotic medications utilized to treat schizophrenia and other related psychiatric disorders. Risperidone primarily acts as an antagonist of dopamine D2 and serotonin 5-HT2A receptors [5], whereas paliperidone, its active metabolite, shares a similar mechanism of action [6]. It is commonly accepted that oral risperidone possesses approximately twice the potency of oral paliperidone [7]. Monitoring the levels of both risperidone and paliperidone in risperidone-treated patients aids clinicians in determining the appropriate dosage. In risperidone recipients, the risperidone to paliperidone ratio serves as an indicator of CYP2D6 enzyme activity [7,8]. Additionally, the concentration-to-dose (C/D) ratio, encompassing both risperidone and paliperidone, serves as a measure of total clearance from the body. Halving the C/D ratio potentially indicates the utilization of CYP3A inducers or non-compliance, whereas an increase potentially suggests a CYP2D6-poor metabolizer phenotype, utilization of CYP2D6 and/or CYP3A4 inhibitors, or potential renal impairment [9]. With regard to vitro studies, they indicate that risperidone and paliperidone exhibit similar receptor binding profiles, with the exception of α1 blockade [10]. However, paliperidone potentially reduces brain penetration as a result of its higher affinity for the transporter P-glycoprotein [11]. The limited available pharmacokinetic data suggest that paliperidone undergoes metabolism through four minor pathways. Unlike risperidone, the clinical relevance of being a CYP2D6 poor metabolizer may not be significant for paliperidone treatment. Information regarding drug interactions involving paliperidone is limited, with renal excretion likely being the primary elimination route [12]. While both medications share common antipsychotic side effects, such as sedation, weight gain, metabolic alterations, and extrapyramidal symptoms (EPS), disparities in pharmacokinetics and receptor binding profiles potentially result in variations with regard to the severity or incidence of specific side effects. Some studies have indicated that paliperidone may confer a lower risk of EPS as opposed to risperidone [13,14]. Paliperidone is available as an oral extended-release formulation for once-daily dosing and intramuscular injections once a month, once every three months, or once every six months, enhancing patient adherence and symptom control stability [15,16]. Risperidone has been commonly prescribed since its introduction, and paliperidone extended-release formulations and intramuscular injections have contributed to its popularity in recent years. Therefore, these medications have gained a significant global market share (about 15 percent of all antipsychotic markets [17]).

Research on TDM and antipsychotic pharmacology has been actively conducted in Europe [6]. Among these, risperidone and paliperidone, which are commonly utilized in patients diagnosed with schizophrenia, have been well studied, with TDM being utilized to explore their pharmacokinetic profiles. According to the Arbeitsgemeinschaft für Neuropsychopharmakologie und Pharmakopsychiatrie Consensus Guidelines [18], the level of recommendation for the TDM with regard to both risperidone and paliperidone is classified as level 2. The level 2 classification indicates that TDM is recommended for dose titration and for special indications or problem solving, and it will also increase the probability of response in non-responders. This level aligns with antipsychotic drugs, such as aripiprazole, ziprasidone, sulpiride, and quetiapine, and is recognized to be re-obtained from plasma concentrations at therapeutically effective doses; it is also related to clinical effects [18]. Despite the necessity and usefulness of TDM for risperidone, it does not appear to be highly or appropriately utilized in clinical settings [19]. Therefore, understanding the status of TDM utilization based on a retrospective study utilizing actual evidence can provide clues for planning prospective or large-scale studies for future TDM applications [20]. However, few retrospective studies have utilized real-world data to explore the effectiveness of TDM in improving clinical symptoms, reducing readmission, and minimizing side effects in patients with schizophrenia with and without TDM. Therefore, the objective of this study was to explore disparities in the characteristics of patients undergoing treatment for schizophrenia by utilizing TDM with risperidone and paliperidone in comparison to without TDM, as well as to evaluate the effectiveness of TDM on clinical outcomes such as hospitalization and compliance in patients with schizophrenia by utilizing a Clinical Data Warehouse (CDW).

## 2. Results

### 2.1. Demographics

The cohort comprised those undergoing TDM (TDM group) and those not undergoing TDM (non-TDM group). The general characteristics, comorbidities, and treatment of the study cohort are exhibited in Table 1. The average age was significantly younger in the TDM group (35.5 years, 41.7 years, *p* = 0.0006), there were fewer drug naïve patients (33.8%, 68.2%, *p* < 0.0001), and there was greater utilization of propranolol (*p* < 0.0001), anticholinergics (*p* < 0.0001), and laxative (*p* = 0.0105) to mitigate side effects in the TDM group. For the subgroup analysis, the TDM group was divided into two groups, namely, within the reference range and over the reference range, in accordance to the risperidone and paliperidone concentrations. The general comorbidities and treatment characteristics of the subgroups are presented in Table 2. There were no significant disparities in characteristics between the two subgroups.

### 2.2. Clinical Outcomes between TDM Group and Non-TDM Group

Treatment period was 864 ± 613 days in TDM group and 772 ± 643 days in non-TDM group (*p* = 0.3019). In the TDM-group, TDM was performed an average of 1.91 times during the treatment period. There were no disparities in hospital stay (0.10 stays/year, 0.10 stays/year, *p* = 0.9082) and hospital days (2.49 days/year, 1.81 days/year, *p* = 0.5861) between the TDM and non-TDM groups. In the TDM group, all hospitalizations occurred after the first TDM (Table 3).

The secondary outcomes are summarized in Table 3. Risperidone and paliperidone doses were higher in the TDM group as opposed to in the non-TDM group (320 mg/day and 252 mg/day, *p* = 0.0045). Additionally, all antipsychotics doses were higher in the TDM group vs. the non TDM group (607 mg/day and 443 mg/day, *p* = 0.0019). Propranolol (37.5%, 21.2%, respectively, *p* = 0.0062) and anticholinergic utilization (66.4%, 53.0%, respectively, *p* = 0.0308) to treat side effects were higher in the TDM group. However, there were no disparities in the utilization of the two drugs when analysis of covariance (ANCOVA) was conducted with regard to age, ratio of drug-naïve patients, propranolol utilization, anticholinergic utilization, and laxative utilization as covariates (*p* = 0.2447 and *p* = 0.6119, respectively). There were no significant disparities in compliance between the two groups (88.6%, 84.1%, *p =* 0.1635). The duration of antidepressant utilization (22.6%, 22.8%, *p* = 0.9701), mood stabilizer utilization (15.7%, 24.6%, *p* = 0.0897), and benzodiazepine utilization (54.5%, 54.5%, *p* = 0.9995) was not significantly different between the two groups. Emergency room visits were not significantly different between the two groups (0.19 visits/year, 0.50 visits/year, *p =* 0.3713).

### 2.3. Clinical Outcome in Subgroups of TDM Group

The treatment period was 938 ± 628 days for those within the reference range group and 743 ± 578 days for those over the reference range (*p* = 0.1700). The subgroups within the TDM group were within the reference range group, and those that were in the over the reference range group, with hospital stays (0.11 stays/year, 0.09 stays/year, *p* = 0.5863) and hospital days (2.36 days/year, 2.70 days/year, *p* = 0.8094), did not differ between the two groups (Table 4). There was no significant disparity in the number of TDM conversions between the two subgroups, and the average concentrations of risperidone and paliperidone obtained post-TDM were higher as opposed to those in the reference range group (29.4 [3.6~49.5] ng/mL, 70.0 [44.4~146.1] ng/mL, *p* < 0.0001). The distribution of mean concentrations of risperidone and paliperidone for each patient within and over the reference range is shown in Figure 1. Of the 50 patients within the reference range group, 9 (18%) had subtherapeutic concentrations with a mean TDM result of less than 20 ng/mL. In two patients in the over the reference range group, the mean concentration exceeded the laboratory alert level—the plasma level at or above which the laboratory should immediately inform the treating physician—of 120 ng/mL. While a patient with a mean concentration of 130.2 ng/mL had no hospitalization, another with a mean concentration of 146.1 ng/mL had hospital stays and hospital days (0.46 stays/year and 5.56 days/year, respectively). Therefore, the clinical outcomes of these two patients were not significantly different from those of other patients in the over the reference range group.

Compliance was higher in the over the reference range group at 94.1% compared to those in the within the reference range group (85.3%; *p* = 0.0206). The doses of all antipsychotics (525 mg/day, 744 mg/day, *p* = 0.0226), and risperidone and paliperidone (280 mg/day, 388 mg/day, *p* = 0.0058) were also higher in the over the reference range group, as was propranolol utilization (29.1%, 51.5%, *p* = 0.0260) and anticholinergic utilization (55.7%, 84.1%, *p* = 0.0025) to treat side effects. The duration of antidepressant utilization (23.1%, 21.8%, *p* = 0.8857) and benzodiazepine utilization (49.1%, 63.5%, *p* = 0.1100) was not significantly different between the two groups. The duration of mood stabilizer utilization (21.9%, 5.3%; *p* = 0.0142) was greater in the within the reference range group. The number of emergency room visits was not significantly different between the two groups (0.20 visits/year, 0.18 visits/year, *p* = 0.8771).

## 3. Discussion

To the best of the researchers’ knowledge, this study represents the initial analysis of the utilization and impacts of risperidone and paliperidone TDM in patients diagnosed with schizophrenia, utilizing real-world data sourced from a CDW. A primary finding of this study revealed that, among patients with schizophrenia treated primarily with risperidone/paliperidone as their primary antipsychotic, those in the TDM group received higher doses of antipsychotics as opposed to those in the non-TDM group. However, despite this discrepancy, no significant difference emerged in one of the most pivotal outcomes of schizophrenia treatment: the ratio of hospitalization and compliance (Table 3).

Prior studies on the TDM of various antipsychotics have exhibited that TDM can enhance treatment efficacy or reduce side effects in patients with psychiatric diseases [21,22,23]. However, to the researchers’ knowledge, minimal studies on antipsychotics have directly compared TDM and non-TDM groups, such as in this study. The findings of this study did not provide significant evidence to support an increase in treatment efficacy or reduction in treatment-related side effects in the TDM group. In contrast to the usual practice of checking drug concentrations and changing the dose through TDM, in this study, the TDM outcome probably helped in deciding whether to adjust the dose; however, it was not the most important factor. Dose adjustment is usually driven by a patient’s response to treatment and adverse events. Therefore, in this study, the physicians seemed to cautiously maintain treatment while monitoring a patient’s adverse events rather than reducing the dose when high concentrations were identified through TDM. Furthermore, patients in the TDM group utilized more therapeutic medications to manage the side effects. This is believed to be a result of the higher frequency of medication utilized to treat side effects, either at the beginning of treatment or when switching to risperidone/paliperidone. This may have also arisen from the finding that patients in the over the range group have higher utilization rates of drugs treating the side effects due to higher doses being given. So, in a way, there is a correlation between doses and higher concentration and incidence of side effects reflected in the use of concomitant medication. As described in the Results section (Table 3), there was no statistically significant disparity in the frequency of drug utilization for managing adverse events between the two groups when demographic disparities such as age, propanol utilization, anticholinergic utilization, and laxative utilization were corrected for covariates in the statistical analysis. 

In the subgroup analysis within the TDM group, the demographic data did not exhibit any statistically significant disparities (Table 2). However, following the administration of risperidone or paliperidone, the utilization of anticholinergics and propranolol was notably higher as opposed to the reference range group (Table 4). These results may stem from the frequent occurrence of side effects caused by high doses of antipsychotics. Interestingly, the utilization of mood stabilizers was lower in the TDM group as opposed to the non-TDM group. Similarly, in the reference range group, the usage frequency of mood stabilizers was lower in the over the reference range group. This could be as a result of patients receiving high doses being expected to have more severe symptoms, leading to a better differentiation between schizophrenia and bipolar disorder. Furthermore, it is recognized that including a mood stabilizer in the treatment of schizophrenia enhances the treatment effect [24]; therefore, clinicians are utilizing this regimen. However, if risperidone or paliperidone is administered at a higher dose through TDM, a combination of mood stabilizers can be avoided. Therefore, it was speculated that the utilization of mood stabilizers decreased in the over the reference range group. In terms of compliance, patients in the over the reference range group exhibited higher compliance, which seems to have occurred as a result of patients receiving high doses requiring more frequent outpatient visits for close follow-up as opposed to those within the reference range group.

Refractory schizophrenia refers to a subtype of schizophrenia in which symptoms persist despite treatment with multiple antipsychotic medications at adequate doses and durations, and it is treated with clozapine [25]. However, clozapine can cause serious side effects, including agranulocytosis and seizures [26,27]. The subgroup analysis of this study (Table 4) exhibited that risperidone and paliperidone were utilized at significantly higher doses in the over the reference range group when compared between the two subgroups, whereas the doses of the other antipsychotics were not statistically different. Nevertheless, there was non-inferiority for compliance and re-hospitalizations in the over the reference range group compared with the within the reference range group. This suggests that higher-than-standard doses of risperidone and paliperidone can be utilized in combination with TDM as an alternative to multiple antipsychotic agents or switching to clozapine in patients with refractory schizophrenia. Reducing the number of medications is expected to reduce side effects by minimizing drug interactions that can occur due to polypharmacy. Although larger studies are needed, the present study suggests the feasibility of higher-than-standard doses of risperidone and paliperidone regimens with TDM for patients with refractory schizophrenia.

A limitation of this study is that TDM was conducted based on clinical practice rather than randomization, which may imply selection bias, as patients with more severe symptoms or poorer treatment responses were more likely to be in the TDM group. Nonetheless, this retrospective study could help us understand the characteristics of TDM and non TDM groups, and the generated data could serve as reference for planning future prospective, randomized studies. Although this real-world-data-based study was limited in its effectiveness compared with a randomized controlled design, it was significant because it was able to identify differences in the characteristics, such as antipsychotic doses for patients who use TDM in clinical practice versus those who do not. Therefore, important insights can be obtained when planning RWD for further studies. Additionally, it is anticipated that these limitations are unlikely to substantially influence the outcomes given that the researchers’ involvement did not extend to the selection of antipsychotics for individual patients. It is noteworthy that the TDM group did not differ significantly from the non-TDM group in hospitalization and complaints, despite the fact that the TDM group may have had poorer symptom severity or treatment response. This finding alone has significant medical value. Secondly, the database provided limited data concerning the severity of symptoms or the occurrence of side effects associated with antipsychotics, such as tardive dyskinesia. Consequently, this information was not utilized as a clinical outcome in this study. In real-world clinical settings, particularly in countries such as Korea, where physicians often handle larger numbers of patients, consistent documentation of patient treatment responses as well as side effects in Electronic Medical Records (EMRs) is exceedingly challenging. In this study, this limitation was addressed by assessing the treatment response of patients with schizophrenia based on antipsychotic dosage, hospitalization, and compliance; side effects were evaluated based on the frequency of therapeutic medication utilization when side effects occurred. Thirdly, this study utilized a CDW based on EMRs from a single hospital. Hence, the complete medical histories of patients spanning their entire lives, such as those contained the National Insurance Claims Database, could not be accessed. Therefore, the medical records of patients who received treatment at different hospitals could not be verified. However, as most institutions tend to record treatments for the same condition during disease management in the EMRs, the EMRs of all patients were reviewed to mitigate this bias. Records of patients receiving schizophrenia treatment at other medical institutions during the treatment period were incorporated into the dataset. Fourthly, this study utilized two antipsychotics, risperidone and paliperidone. Some patients took only one drug, whereas others utilized both drugs. During the observation period, the drug regimen was changed for each patient. In addition, risperidone has numerous similarities and disparities from paliperidone; therefore, the possibility that these factors influenced the results of this study could not be ruled out. In future, a large-scale study that controls for these variables is required. Finally, this study utilized a relatively small sample of approximately 200 patients, and the dataset did not include pharmacogenetic information related to drug metabolism and action, such as CYP2D6 enzyme measurements. A larger sample combined with pharmacogenetic information would allow for a more precise study. These limitations can be overcome by further studies that utilize CDWs where multiple healthcare organizations share EMRs and collect pharmacogenetic information.

## 4. Materials and Methods

### 4.1. CDW and Data Collection

This was a real-world evidence study based on retrospective medical records retrieved from the CDW of a single center—the CHA Bundang Medical Center— focusing on the utilization of risperidone or paliperidone in patients diagnosed with schizophrenia. A CDW is a data platform that collects and distributes clinical data to researchers, including more than eight million EMRs, such as diagnoses, medications, hospitalizations, and procedures, that are completely anonymized, with access being granted only for research purposes, post a comprehensive ethical board review. This study was conducted in accordance with the Declaration of Helsinki and approved by the Institutional Review Board of CHA University, CHA Bundang Medical Center, Seongnam, Korea (IRB No.: CHAMC 2023-05-010 and date of approval: 8 May 2023). As this study used a CDW, compliance with the Personal Information Protection Act of South Korea was ensured.

Schizophrenia and comorbidities were identified utilizing the International Classification of Diseases, 10th Revision, while prior and concurrent medications, including antipsychotics, were extracted utilizing the ingredient name of the Anatomical Therapeutic Chemical classification system. Patient visits included hospitalizations under psychiatric care, outpatient consultations, and emergency room visits for psychiatric diseases. Medication data were extracted for drug names, ingredient names, prescription duration, and dosing regimens, with groupings for analysis including antipsychotics (risperidone, paliperidone, amisulpride, aripiprazole, chlorpromazine, haloperidol, olanzapine, quetiapine, and ziprasidone), antidepressants (desvenlafaxine, duloxetine, escitalopram, fluoxetine, fluvoxamine, mirtazapine, paroxetine, sertraline, venlafaxine, and vortioxetine), mood stabilizers (carbamazepine, lamotrigine, lithium, topiramate, and valproic acid), benzodiazepines (alprazolam, bromazepam, clonazepam, diazepam, flurazepam, and lorazepam), propranolol, anticholinergics (benztropine and trihexyphenidyl), and laxatives (bisacodyl, lactulose, magnesium oxide, and prucalopride).

Data were collected to check whether the patients were treated with other antipsychotics prior to treatment with risperidone and paliperidone. Patients were defined as drug-naïve if they had not taken antipsychotics within seven days prior to treatment with risperidone or paliperidone. Propranolol, anticholinergics, and laxative prescription data were collected up to 30 days prior to treatment to validate whether medications were utilized to mitigate side effects prior to the beginning of the treatments.

### 4.2. Study Population

Data were collected from 1481 patients who were diagnosed with schizophrenia (codes F20 and F21 as principal diagnoses) between 1 November 2017 and 31 October 2023. A total of 707 patients visited the hospital at least once and underwent treatment with risperidone or paliperidone during the above period; 412 patients did not utilize risperidone, paliperidone, or clozapine from 1 January 2015 to 31 December 2016, excluding patients resistant to schizophrenia treatment and patients who started risperidone and paliperidone treatment prior to the TDM set-up in the study’s primary hospital. The number of patients who were aged between 18 and 65 years was 378. Elderly and pediatric patients were excluded as they may have clinical treatment response and pharmacokinetic differences. Patients with follow-up periods of 12 weeks or less were excluded because they were observed for too short a period, and a total of 212 patients were followed up for 12 weeks post the first administration of risperidone or paliperidone. Among these, 80 patients underwent TDM with risperidone or paliperidone at least once during the treatment period (TDM group), and 132 patients did not undergo TDM during the treatment period (non-TDM group). The proportion of toxic levels among all the TDM cases for each patient was calculated by reviewing all the TDM cases for each patient in the TDM group. Fifty patients had risperidone or paliperidone concentrations within the reference range in over 50% of all TDM (within the reference group). Thirty patients had risperidone or paliperidone concentrations that were over the reference range by more than 50% in all TDM (over the reference group). A flowchart of the study population is exhibited in Figure 2.

### 4.3. Evaluation of Primary Outcomes

The primary outcomes encompassed the quantity and cumulative duration of psychiatric hospitalizations. Cumulative psychiatric hospitalization duration was computed by aggregating the lengths of all psychiatric hospital stays. Psychiatric hospitalization was delineated as an inpatient residency in the psychiatric ward lasting a minimum of two days. These primary outcomes were evaluated throughout the treatment phase, during which the patient underwent risperidone or paliperidone therapy, and was subsequently normalized over the total treatment period (in years).

### 4.4. Evaluation of Secondary Outcomes

#### 4.4.1. Treatment Period 

The treatment period was from the date of the first dose of risperidone or paliperidone (first prescription) to the date of the last dose (last prescription plus the number of days of treatment).

#### 4.4.2. Compliance 

Compliance was assessed through the calculation of the proportion of days covered (PDC) by the patient for antipsychotics. This metric was derived by dividing the total number of treatment days with antipsychotics by the overall number of treatment days. Notably, for paliperidone, long-acting injectable (LAI) formulations were administered at monthly and three-month intervals, and the treatment duration was standardized to 28 days and 84 days, respectively. Charts of each patient by each visit were reviewed, and if the patient’s primary care physician stated that the patient did not take their medication, they were excluded from the analysis.

#### 4.4.3. Antipsychotics Dose and Concomitant Medications

All antipsychotic doses were standardized to equivalent doses of chlorpromazine. The cumulative doses of all antipsychotics administered to each patient were aggregated and then divided by the total number of treatment days.

Concomitant medication utilization was assessed using the PDC metric. The duration of concomitant medication usage was computed by dividing the total number of days of concomitant medication treatment by the total number of days of the prescribed treatment period.

#### 4.4.4. Emergency Visit

Data were only collected when schizophrenia was the primary diagnosis. These outcomes were included for each patient and divided by the total number of years with regard to treatment.

### 4.5. Therapeutic Drug Monitoring

Patients in the TDM group underwent blood sampling to measure the risperidone and paliperidone concentrations. Trough sampling was performed, and blood was collected whenever possible after a maintenance dose was reached at a steady state. The blood samples used to test the concentrations of risperidone and paliperidone were subjected to standard protein precipitation utilizing a high-pressure liquid chromatography/mass spectrometry (LC-MS/MS) analyzer. The lower limit of quantification was 0.5 ng/mL for both risperidone and paliperidone (9-hydroxyrisperidon). The findings of the concentration were utilized to estimate the peak and trough concentrations of the medications utilizing established pharmacokinetic models (for oral formulations of risperidone [28], paliperidone [29], and paliperidone LAI formulations administered at monthly [30] as well as three-month [31] intervals). Considering that the combined reference range for the known concentrations of risperidone and paliperidone was 20–60 ng/mL [2], the findings of each TDM test were interpreted and classified on a case-by-case basis into therapeutic, subtherapeutic, and toxic levels. The laboratory alert level was 120 ng/mL [2]. The number of TDM consultations for each patient and the average concentrations of risperidone and paliperidone at the time of blood collection were collected based on clinical outcomes.

### 4.6. Statistical Analysis

Statistical analysis was conducted from 1 November 2017 to 31 October 2023. Group comparisons of demographic characteristics and clinical outcomes were conducted utilizing the chi-square test or Fisher’s exact test for categorical variables and independent *t*-tests for continuous variables. Additionally, ANCOVA was utilized to compare primary and secondary outcomes between groups (i.e., the number of hospitalization stays, total length of hospitalization stays, treatment compliance, dose of antipsychotics, etc.) while controlling for observed disparities between groups for covariates that exhibited statistically significant disparities between groups among all the above-mentioned covariates. The TDM and non-TDM groups were compared, and the within and over the reference range groups were compared utilizing subgroup analysis. All statistical analyses were conducted utilizing SPSS version 29 (IBM Corporation, Armonk, NY, USA), and a 2-tailed level of statistical significance was defined as *p* < 0.05.

Multiple linear regression was conducted to obtain the relevant variables for ANCOVA. The following variables were considered independent variables, namely, sex, age, psychiatric comorbidities, and concomitant medication with regard to side effects (such as anticholinergics, propranolol, and laxatives). Comorbidity was characterized by the inclusion of any additional diagnostic codes indicating conditions such as depressive disorders (codes F32, F33, and F34), bipolar disorders (codes F30 and F31), anxiety disorders (codes F40 and F41), and substance use disorders (codes F10–F19) along with a primary diagnosis of schizophrenia.

## 5. Conclusions

TDM of risperidone and paliperidone was predominantly employed for patients with schizophrenia who received high doses and exhibited either poor treatment responses or anticipated significant side effects. However, upon comparison with patients who did not undergo TDM, no significant disparities were observed in critical outcomes such as hospitalization rates, treatment adherence, and visits to the emergency room for schizophrenia treatment. In summary, this study provides evidence that TDM of risperidone and paliperidone could maintain patient stability, even when high doses are administered to high-risk individuals with schizophrenia.

## Figures and Tables

**Figure 1 pharmaceuticals-17-00882-f001:**
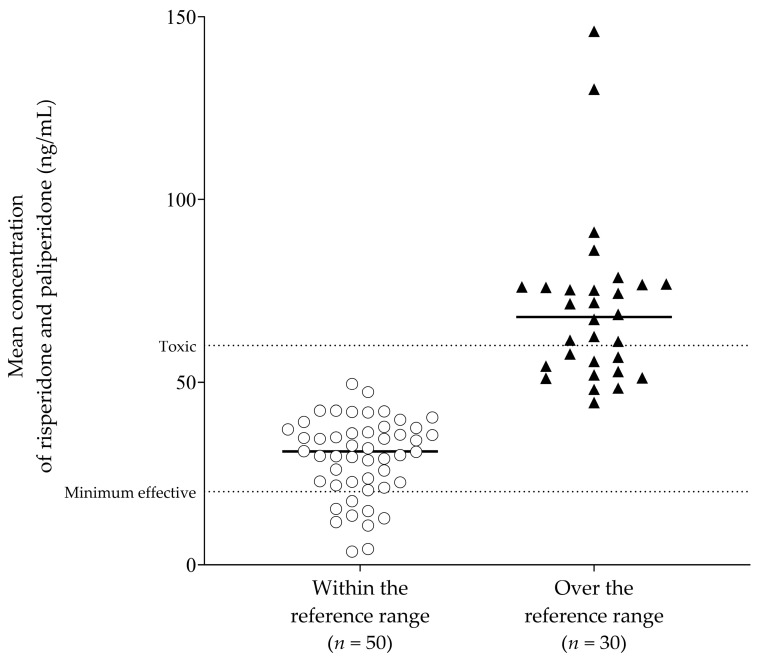
The distribution of mean concentrations of risperidone and paliperidone for each patient in the within and the over the reference range groups.

**Figure 2 pharmaceuticals-17-00882-f002:**
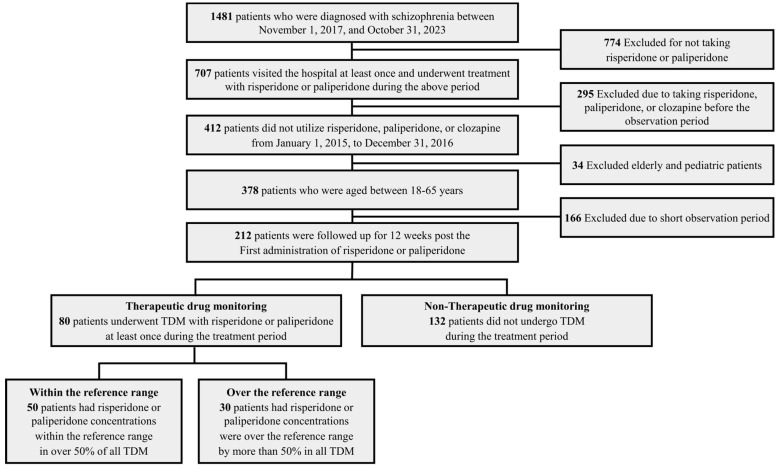
Flowchart exhibiting the study population.

**Table 1 pharmaceuticals-17-00882-t001:** Demographics and clinical characteristics in each treatment group.

Characteristics	TDM (n = 80)	Non-TDM (n = 132)	*p*-Value
Age (years)	35.5 (11.5)	41.7 (13.1)	0.0006
Male (%)	35.0	37.9	0.6735
Comorbidity			
Depression (%)	21.3	25.0	0.5330
Bipolar disorder (%)	22.5	18.9	0.5320
Anxiety (%)	45.0	37.1	0.2565
Substance use (%)	1.3	4.5	0.2583 ^1^
Drug-naïve patient (%)	33.8	68.2	<0.0001
Prior treatment period with antipsychotics prior to utilizing risperidone and paliperidone (days)	581.9 (1338.8)	738.7 (1472.3)	0.4378
Medication to mitigate side effects in prior treatments			
Propranolol utilization (%)	31.3	6.1	<0.0001
Anticholinergics utilization (%)	47.5	18.2	<0.0001
Laxative utilization (%)	16.3	6.1	0.0105

^1^ Fisher’s exact test was utilized. Numeric data are presented as mean (standard deviation).

**Table 2 pharmaceuticals-17-00882-t002:** Demographics and clinical characteristics in each subgroup of therapeutic drug monitoring group.

Characteristics	Within the Reference Range (*n* = 50)	Over the Reference Range (*n* = 30)	*p*-Value
Age (years)	34.4 (10.4)	37.2 (13.1)	0.2890
Male (%)	38.0	30.0	0.4677
Comorbidity			
Depression (%)	26.0	13.3	0.2604 ^1^
Bipolar disorder (%)	24.0	20.0	0.6783
Anxiety (%)	40.0	53.3	0.2458
Substance use (%)	2.0	0.0	1.0000 ^1^
Drug-naïve patient (%)	38.0	26.7	0.2993
Prior treatment period with antipsychotics prior to utilizing risperidone and paliperidone (days)	529.0 (1240.7)	670.0 (1506.2)	0.6513
Medication to mitigate side effects in prior treatments			
Propranolol utilization (%)	26.0	40.0	0.1909
Anticholinergics utilization (%)	44.0	53.3	0.4183
Laxative utilization (%)	16.0	16.7	0.9376

^1^ Fisher’s exact test was utilized. Numeric data are presented as mean (standard deviation).

**Table 3 pharmaceuticals-17-00882-t003:** Clinical outcomes in each treatment group.

Characteristics	TDM (*n* = 80)	Non-TDM (*n* = 132)	*p*-Value ^2^	*p*-Value ^2,3^ (ANCOVA)
Treatment period (days)	864 (613)	772 (643)	0.3019	0.8302
Hospital stays (stays/year)	0.10 (0.21)	0.10 (0.47)	0.9082	0.5502
Hospital days (days/year)	2.49 (6.03)	1.81 (10.13)	0.5861	0.3125
Emergency room visits (visits/year)	0.19 (0.56)	0.50 (3.03)	0.3713	0.6388
Compliance (%)	88.6 (20.2)	84.1 (24.0)	0.1635	0.7516
All antipsychotics dose (mg/day) ^1^	607 (393)	443 (353)	0.0019	0.0538
Risperidone and paliperidone dose (mg/day) ^1^	320 (172)	252 (165)	0.0045	0.0082
Other antipsychotics dose (mg/day) ^1^	287 (328)	191 (289)	0.0272	0.3934
Duration of Antidepressant utilization (%)	22.6 (38.2)	22.8 (39.7)	0.9701	0.7920
Duration of Mood stabilizer utilization (%)	15.7 (34.3)	24.6 (40.7)	0.0897	0.0147
Duration of Benzodiazepine utilization (%)	54.5 (39.1)	54.5 (43.5)	0.9995	0.8903
Duration of Propranolol utilization (%)	37.5 (43.7)	21.2 (37.5)	0.0062	0.2447
Duration of Anticholinergics utilization (%)	66.4 (41.5)	53.0 (46.1)	0.0308	0.6119
Duration of Laxative utilization (%)	16.5 (32.1)	9.0 (24.6)	0.0746	0.0598
Average concentration of risperidone and paliperidone (ng/mL)	44.6 (25.4)			
Number of TDM consultation (number/year)	1.91 (2.08)			

^1^ Antipsychotics doses were converted to chlorpromazine equivalents. ^2^ Group comparisons were conducted utilizing the chi-square test or Fisher’s exact test for categorical variables and independent t-tests for continuous variables. ^3^ Covariates: age and propranolol, anticholinergic, laxative utilization in Table 1. Data are presented as mean (standard deviation).

**Table 4 pharmaceuticals-17-00882-t004:** Clinical outcomes in each subgroup of the therapeutic drug monitoring group.

Characteristics	Within the Reference Range (*n* = 50)	Over the Reference Range (*n* = 30)	*p*-Value ^2^
Average concentration of risperidone and paliperidone (ng/mL)	29.4 (10.9)	70.0 (22.3)	<0.0001
Number of TDM consultations (number/year)	1.60 (1.88)	2.43 (2.31)	0.0853
Treatment period (days)	938 (628)	743 (578)	0.1700
Hospital stays (stays/year)	0.11 (0.22)	0.09 (0.20)	0.5863
Hospital days (days/year)	2.36 (5.04)	2.70 (7.49)	0.8094
Emergency room visits (visits/year)	0.20 (0.54)	0.18 (0.60)	0.8771
Compliance (%)	85.3 (24.2)	94.1 (8.0)	0.0206
All antipsychotics dose (mg/day) ^1^	525 (343)	744 (436)	0.0226
Risperidone and paliperidone dose (mg/day) ^1^	280 (148)	388 (189)	0.0058
Other antipsychotics dose (mg/day) ^1^	245 (290)	356 (378)	0.1713
Duration of Antidepressant utilization (%)	23.1 (38.0)	21.8 (39.2)	0.8857
Duration of Mood stabilizer utilization (%)	21.9 (39.8)	5.3 (19.0)	0.0142
Duration of Benzodiazepine utilization (%)	49.1 (39.9)	63.5 (36.6)	0.1100
Duration of Propranolol utilization (%)	29.1 (41.4)	51.5 (44.7)	0.0260
Duration of Anticholinergics utilization (%)	55.7 (43.9)	84.1 (30.3)	0.0025
Duration of Laxative utilization (%)	16.4 (33.6)	16.7 (30.1)	0.9698

^1^ Antipsychotics doses were converted to chlorpromazine equivalents. ^2^ Group comparisons were conducted utilizing the chi-square test or Fisher’s exact test for categorical variables and independent t-tests for continuous variables. Data are presented as mean (standard deviation).

## Data Availability

The data are not publicly available due to confidentiality reasons.

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
