# Peer review of "Assessing the Clinical Efficacy of Therapeutic Drug Monitoring for Risperidone and Paliperidone in Patients with Schizophrenia: Insights from a Clinical Data Warehouse"

_pharmaceuticals, 2024, doi:10.3390/ph17070882_

Round 1

Reviewer 1 Report

Comments and Suggestions for Authors

As mentioned by the authors, there was no randomisation between TDM and non TDM patients. It shoud be discussed if patients in more serious conditions are more liekely to receive TDM introducing a selection bias. 

Were there no patients in TDM group with subtherapeutic concentrations identified? This is also be relevant for assessing the value of TDM and information on this must be added..

Line 220: what exactly is the new treatment regimen proposed?

Line 300: how are toxic levels defined? I understand form line 349 that the theraeutic range was 20-60 ng/ml risperidone and paliperidone (sum of both?)

Line346: the limit of quantification must be reported.

Overall, some more information on the TDM results, i.e. concentration range etc. should be reported.

Comments on the Quality of English Language

Please check for inconsistencies, for example line 87: what does "and personolized medicine" mean here?

Author Response

Reviewer #1:

Comment 1: As mentioned by the authors, there was no randomisation between TDM and non TDM patients. It shoud be discussed if patients in more serious conditions are more liekely to receive TDM introducing a selection bias.

Response 1: Thank you for reviewing our manuscript and providing insightful and helpful comments. As you pointed out, this study was not randomized, suggesting the possibility of a selection bias resulting in patients with more severe conditions being assigned to the TDM group. We have addressed this as a limitation of the study in the Discussion section as follows:

Lines 250–255:

“A limitation of this study is that TDM was conducted based on clinical practice rather than randomization, which may imply selection bias as patients with more severe symptoms or poorer treatment responses were more likely to be in the TDM group. Nonetheless, this retrospective study could help us understand the characteristics of TDM and non TDM groups, and the generated data could serve as reference for planning future prospective, randomized studies.”

Comment 2: Were there no patients in TDM group with subtherapeutic concentrations identified? This is also be relevant for assessing the value of TDM and information on this must be added..

Response 2: Of the 50 patients in the within the reference range group, 9 (18%) had subtherapeutic concentrations with a mean TDM result of less than 20 ng/mL. However, as TDM was usually conducted during the treatment period with a maintenance dose, these patients were treated with a sufficient dose rather than a lower one to achieve a therapeutic effect. We have added text to the Methods and Results sections accordingly.

Lines 149–163:

“There was no significant disparity in the number of TDM conversions between the two subgroups, and the average concentrations of risperidone and paliperidone obtained post TDM were higher as opposed to those in the reference range group (29.4 [3.6 ~ 49.5] ng/mL, 70.0 [44.4 ~146.1] ng/mL, p < 0.0001). The distribution of mean concentrations of risperidone and paliperidone for each patient in the within and over the reference range groups is shown in Figure 1. Of the 50 patients in the within the reference range group, 9 (18%) had subtherapeutic concentrations with a mean TDM result of less than 20 ng/mL. In two patients in the over the reference range group, the mean concentration exceeded the laboratory alert level—the plasma level at or above which the laboratory should immediately inform the treating physician—of 120 ng/mL. While a patient with a mean concentration of 130.2 ng/mL had no hospitalization, another with a mean concentration of 146.1 ng/mL had hospital stays and hospital days (0.463 stays/year and 5.558 days/year, respectively). Therefore, the clinical outcomes of these two patients were not significantly different from those of other patients in the over the reference range group.”

Lines 385–387:

“Patients in the TDM group underwent blood sampling to measure risperidone and paliperidone concentrations. Trough sampling was performed; blood was collected whenever possible after maintenance dose was reached.”

Comment 3: Line 220: what exactly is the new treatment regimen proposed?

Response 3: Under the new treatment regimen, we propose high-dose risperidone and paliperidone, not multiple antipsychotic agents or switching to clozapine, as is usually done for patients with refractory schizophrenia. Reducing the number of medications is expected to reduce side effects by minimizing drug interactions that can occur due to polypharmacy. This has been addressed in the Discussion section.

Lines 242–249:

”This suggests that higher-than-standard doses of risperidone and paliperidone can be utilized in combination with TDM as an alternative to multiple antipsychotic agents or switching to clozapine in patients with refractory schizophrenia. Reducing the number of medications is expected to reduce side effects by minimizing drug interactions that can occur due to polypharmacy. Although larger studies are needed, the present study suggests the feasibility of higher-than-standard doses of risperidone and paliperidone regimens with TDM for patients with refractory schizophrenia.”

Comment 4: Line 300: how are toxic levels defined? I understand form line 349 that the theraeutic range was 20-60 ng/ml risperidone and paliperidone (sum of both?).

Response 4: Toxic levels (laboratory alert level) are defined as >120 ng/mL. Of the 30 patients in the over the reference range group, 2 (6.7%) were in this category. The primary outcomes for these patients were 130.2 ng/mL and 146.1 ng/mL, respectively, and they were not outliers in the over the reference range group. The patient with a mean concentration of 130.2 ng/mL had no hospitalizations, and the patient with a mean concentration of 146.1 ng/mL had hospital stays and hospital days of 0.463 stays/year and 5.558 days/year, respectively. Toxicity levels were further described in Methods and Results sections.

Lines 149–163:

“There was no significant disparity in the number of TDM conversions between the two subgroups, and the average concentrations of risperidone and paliperidone obtained post TDM were higher as opposed to those in the reference range group (29.4 [3.6 ~ 49.5] ng/mL, 70.0 [44.4 ~146.1] ng/mL, p < 0.0001). The distribution of mean concentrations of risperidone and paliperidone for each patient in the within and over the reference range groups is shown in Figure 1. Of the 50 patients in the within the reference range group, 9 (18%) had subtherapeutic concentrations with a mean TDM result of less than 20 ng/mL. In two patients in the over the reference range group, the mean concentration exceeded the laboratory alert level—the plasma level at or above which the laboratory should immediately inform the treating physician—of 120 ng/mL. While a patient with a mean concentration of 130.2 ng/mL had no hospitalization, another with a mean concentration of 146.1 ng/mL had hospital stays and hospital days (0.463 stays/year and 5.558 days/year, respectively). Therefore, the clinical outcomes of these two patients were not significantly different from those of other patients in the over the reference range group.”

Lines 397–398:

“The laboratory alert level was 120 ng/mL [2].”

Comment 5: Line346: the limit of quantification must be reported.

Response 5: The lower limit of quantification (LLOQ) is 0.5 ng/mL for both risperidone and paliperidone (9-hydroxyrisperidon). This information has been added in the Methods section.

Lines: 387-391:

“The blood samples for the concentration of risperidone and paliperidone were subjected to standard protein precipitation utilizing a high-pressure liquid chromatography/mass spectrometry (LC-MS/MS) analyzer. The lower limit of quantification is 0.5 ng/mL for both risperidone and paliperidone (9-hydroxyrisperidon).”

Comment 6: Overall, some more information on the TDM results, i.e. concentration range etc. should be reported.

Response 6: We have described the distribution of mean concentrations of risperidone and paliperidone for each patient in the within and over the reference range groups in Figure 1 (new Figure). We have also added the concentration range for each group in the Results section.

Lines 149–154:

“There was no significant disparity in the number of TDM conversions between the two subgroups, and the average concentrations of risperidone and paliperidone obtained post TDM were higher as opposed to those in the reference range group (29.4 [3.6 ~ 49.5] ng/mL, 70.0 [44.4 ~146.1] ng/mL, p < 0.0001). The distribution of mean concentrations of risperidone and paliperidone for each patient in the within and over the reference range groups is shown in Figure 1.”

Comment 7: Please check for inconsistencies, for example line 87: what does "and personolized medicine" mean here? (Comments on the Quality of English Language)

Response 7: Based on your comments, we have identified and corrected inconsistencies in the manuscript. The phrase “and personalized medicine” has been removed from the sentence to avoid confusion.

Reviewer 2 Report

Comments and Suggestions for Authors

The researcher performed a retrospective study of TDM vs non-TDM of risperidon and paliperidone from retrospective data from a single center. From a clinical dataset they  screened the full medicaiton record of 1481 patinets diagnosed with schizofrenia, of whom 212 patinets received risperiodone or paliperidone of which 132 not combined with TDM and  80 in combination with TDM and analysed the observed concentrations, dose, hospitalization, concomitant medication and other factors. I have the following suggestions.

The abstract is not informative. Please add the number of patients that were selected and main outcomes (Table3 and 4) including Pvalues in the abstract.

The term Clinical Data Warehouse remains unclear, specifically in the abstract. Please describe that retrospective data were analysed from a single center and provide details of the hospital both in the abstract and main text.

Methods: Could the authors report the ethical approval number of the study. I wonder how anonimized the dataset was as information about diagnosis, treatment, hospital stays etc was collected. Was the study approved accorinding to GDPR?

Can the authors asses the effect of TDM on dose utilization? Were doses adjusted after TDM

The compliace estimate is unclear: how do the researcher know whether patients actually took the drug during the treatment period? Please explain.

Was pharmacogenetic information (eg CYP2D6 expression) available for any of these patients? If not discuss this as a limititation.

In Table 3 and Table 4 it remains unclear what the timing is: were these results (dose adjustments/hospitilization etc) as a result/after TDM or at the same time. Please explain and preferably present both.

The results of TDM and dose average TDM results are currenty lacking and are of interest to show.

As in only a low number of patients TDM was performed, based on a clinical decsion, this  likely biased the results described in table 3. Potentially, the physician decised to measure the concentration after the observation of toxicity or nonreponse at standard/highest dosing. Please reflect on this bias, both in abstract and discussion.

Author Response

Reviewer #2:

The researcher performed a retrospective study of TDM vs non-TDM of risperidon and paliperidone from retrospective data from a single center. From a clinical dataset they  screened the full medicaiton record of 1481 patinets diagnosed with schizofrenia, of whom 212 patinets received risperiodone or paliperidone of which 132 not combined with TDM and 80 in combination with TDM and analysed the observed concentrations, dose, hospitalization, concomitant medication and other factors. I have the following suggestions.

Thank you for reviewing our manuscript and providing insightful and helpful comments.

Comment 1: The abstract is not informative. Please add the number of patients that were selected and main outcomes (Table3 and 4) including P-values in the abstract..

Response 1: We have revised the abstract as per your suggestion. However, given the 200-word limit for the abstract, we weren’t able make substantial revisions.

Lines 23–27:

“The findings revealed that patients in the TDM group received higher risperidone and paliperidone doses (320 mg/day and 252 mg/day, p = 0.0045) compared to their non-TDM counterparts. Nevertheless, significant disparities were observed in hospitalization rates, duration of hospital stays, or compliance between the two groups (p = 0.9082, 0.5861, 0.7516, respectively).”

Comment 2: The term Clinical Data Warehouse remains unclear, specifically in the abstract. Please describe that retrospective data were analysed from a single center and provide details of the hospital both in the abstract and main text.

Response 2: In the abstract, we have indicated that “retrospective” data from a “single center” were analyzed. For CDW, we have provided details in the Methods section.

Lines 16–18:

“This study investigated the usage patterns and impact of Therapeutic Drug Monitoring (TDM) for risperidone and paliperidone in patients diagnosed with schizophrenia, utilizing retrospective real-world data sourced from a single center’s Clinical Data Warehouse.”

Lines 294–296:

“This was a post-hoc study based on retrospective medical records retrieved from the CDW of a single center—the CHA Bundang Medical Center—, focusing on the utilization of risperidone or paliperidone in patients diagnosed with schizophrenia.”

Comment 3: Methods: Could the authors report the ethical approval number of the study. I wonder how anonimized the dataset was as information about diagnosis, treatment, hospital stays etc was collected. Was the study approved accorinding to GDPR?

Response 3: The ethical approval number of the study is described in the Institutional Review Board Statement (Lines 438–441). Moreover, as this study uses CDW, we ensured compliance with the Personal Information Protection Act of South Korea. We have added this information in the Methods section.

Lines 296–304:

“A CDW is a data platform that collects and distributes clinical data to researchers, including more than eight million EMRs, such as diagnoses, medications, hospitalizations, and procedures, that are completely anonymized with access granted only for research purposes, post a comprehensive ethical board review. This study was conducted in accordance with the Declaration of Helsinki, and approved by the Institutional Review Board of CHA University, CHA Bundang Medical Center, Seongnam, Korea (IRB No.: CHAMC 2023-05-010 and date of approval: 08-May-2023). As this study used a CDW, compliance with the Personal Information Protection Act of South Korea was ensured.”

Comment 4: Can the authors assess the effect of TDM on dose utilization? Were doses adjusted after TDM. 

Response 4: In this study, the outcome of TDM helped in deciding whether to adjust the dose, but it was not the most important factor; dose adjustment is mainly driven by the patient’s response to treatment and adverse events. This information has been added in the Discussion section.

Lines 195–203:

“The findings of this study did not provide significant evidence to support an increase in treatment efficacy or reduction in treatment-related side effects in the TDM group. In contrast to the usual practice of checking drug concentrations and changing the dose through TDM, in this study, the TDM outcome helped in deciding whether to adjust the dose; however, it was not the most important factor. Dose adjustment is driven by a patient’s response to treatment and adverse events. Therefore, in this study, the physicians cautiously maintained treatment while monitoring a patient’s adverse events rather than reducing the dose when high concentrations were identified through TDM.”

Comment 5: The compliance estimate is unclear: how do the researcher know whether patients actually took the drug during the treatment period? Please explain.

Response 5: This is a limitation of using real world data. We reviewed charts of each patient and excluded those from the analysis if the patient's primary care physician stated that the patient did not take their medication. If the patient lied or the primary care physician did not chart the patient’s non-adherence, the data could be inaccurate. We have addressed this in the Methods section.

Lines 365–369:

“Notably, for paliperidone long-acting injectable (LAI) formulations administered at monthly and three-month intervals, the treatment duration was standardized to 28 days and 84 days, respectively. Charts of each patient by each visit were reviewed and if the patient’s primary care physician stated that the patient did not take their medication, they were excluded from the analysis.”

Comment 6: Was pharmacogenetic information (eg CYP2D6 expression) available for any of these patients? If not discuss this as a limititation.

Response 6: Pharmacogenetic information such as CYP2D6 was not available for the patients in this study; this limitation was discussed in the Discussion section.

Lines 286–291:

“Finally, this study utilized a relatively small sample of approximately 200 patients, and the dataset did not include pharmacogenetic information related to drug metabolism and action, such as CYP2D6. A larger sample combined with pharmacogenetic information would allow for a more precise study. These limitations can be overcome by further studies that utilize CDWs where multiple healthcare organizations share EMRs and collect pharmacogenetic information.”

Comment 7 In Table 3 and Table 4 it remains unclear what the timing is: were these results (dose adjustments/hospitilization etc) as a result/after TDM or at the same time. Please explain and preferably present both.

Response 7: In this study, the observation period (treatment period) is from the date of the first dose of risperidone or paliperidone (first prescription) to the date of the last dose (last prescription plus the number of days of treatment). In the TDM group, all hospitalizations occurred after the first TDM. The above is described in the Methods and Results sections.

Dose adjustments or changes in regimen were not significantly associated with the date of TDM initiation, as these are influenced by the physician’s judgment rather than TDM results.

Lines 123–126:

“There were no disparities in hospital stay (0.103 stays/year, 0.097 stays/year, p = 0.9082) and hospital days (2.489 days/year, 1.808 days/year, p = 0.5861) between the TDM and non-TDM groups. In the TDM group, all hospitalizations occurred after the first TDM (Table 3).”

Lines 356–359:

“4.4.1. Treatment period

The treatment period was from the date of the first dose of risperidone or paliperidone (first prescription) to the date of the last dose (last prescription plus the number of days of treatment).”

Comment 8 The results of TDM and dose average TDM results are currenty lacking and are of interest to show.

Response 8: We have described the distribution of mean concentrations of risperidone and paliperidone for each patient in the within and over the reference range groups in Figure 1 (new Figure). We have also added the concentration range for each group in the Results section.

Lines 149–154:

“There was no significant disparity in the number of TDM conversions between the two subgroups, and the average concentrations of risperidone and paliperidone obtained post TDM were higher as opposed to those in the reference range group (29.4 [3.6 ~ 49.5] ng/mL, 70.0 [44.4 ~146.1] ng/mL, p < 0.0001). The distribution of mean concentrations of risperidone and paliperidone for each patient in the within and over the reference range groups is shown in Figure 1.”

Comment 9: As in only a low number of patients TDM was performed, based on a clinical decsion, this likely biased the results described in table 3. Potentially, the physician decised to measure the concentration after the observation of toxicity or nonreponse at standard/highest dosing. Please reflect on this bias, both in abstract and discussion.

Response 9: As you point out, the study was not randomized and had a small sample size, suggesting the possibility of a selection bias, resulting in patients with more severe conditions being assigned to the TDM group. We have addressed this limitation in the Discussion section, as well as in the abstract briefly.

Lines 250–255:

“A limitation of this study is that TDM was conducted based on clinical practice rather than randomization, which may imply selection bias as patients with more severe symptoms or poorer treatment responses were more likely to be in the TDM group. Nonetheless, this retrospective study could help us understand the characteristics of TDM and non TDM groups, and the generated data could serve as reference for planning future prospective, randomized studies.”

Lines 29–31:

“Despite the possibility of a selection bias in assigning patients to the groups, this study provides a comprehensive analysis of TDM utilization and its ramifications on schizophrenia treatment outcomes.”

Round 2

Reviewer 1 Report

Comments and Suggestions for Authors

Thank you for adressing my comments

Author Response

I saw that there are no additional comments from the reviewer, and I submit the reply.

Reviewer 2 Report

Comments and Suggestions for Authors

The authors responded adequate to all questions and improved the manuscript. I have no further questions 

Author Response

(The authors gave the same response as above.)
